# Antioxidant, Anti-Diabetic, Anti-Obesity, and Antihypertensive Properties of Protein Hydrolysate and Peptide Fractions from Black Sesame Cake

**DOI:** 10.3390/molecules28010211

**Published:** 2022-12-26

**Authors:** Supakit Chaipoot, Wanisa Punfa, Sakaewan Ounjaijean, Rewat Phongphisutthinant, Kanokwan Kulprachakarn, Wason Parklak, Laddawan Phaworn, Pattrawan Rotphet, Kongsak Boonyapranai

**Affiliations:** 1Science and Technology Research Institute, Chiang Mai University, Chiang Mai 50200, Thailand; 2School of Traditional and Alternative Medicine, Chiang Rai Rajabhat University, Chiang Rai 57100, Thailand; 3Research Institute for Health Sciences, Chiang Mai University, Chiang Mai 50200, Thailand; 4Faculty of Allied Health Sciences, Nakhon Ratchasima College, Nakhon Ratchasima 30000, Thailand

**Keywords:** in vitro study, sesame, enzymatic hydrolysis, peptide, membrane filtration

## Abstract

A low-value by-product of cold-pressed sesame oil is defatted black sesame cake (DBSC). The remaining protein and essential amino acids may be utilized as a renewable biological source to produce bioactive products. The bioactivities of the protein hydrolysate from black sesame cake and its peptide fractions were examined in this study for in vitro antioxidant activity and inhibition of DPP-IV, ACE, α-amylase, α-glucosidase, and pancreatic lipase. By using Flavourzyme to hydrolyze DBSC, followed by ultrafiltration, fractions with peptide sizes of <3, 3–10, and >10 kDa were obtained. According to the findings, the products of DBSC could neutralize free radicals and prevent ferric ion redox reactions. The highest inhibitory effects were shown with low Mw peptides (<3 kDa) against ACE, DPP-IV, α-amylase, and α-glucosidase. DBSC has demonstrated potential as a nutraceutical or functional ingredient for preventing and treating disorders associated with free radicals, such as diabetes, hypertension, and hyperglycemia.

## 1. Introduction

Enzymatic hydrolysis is useful for enhancing the nutritional and functional qualities of protein-rich foods. Better solubility, foaming, emulsifying properties, and the release of physiologically active peptides from proteins have benefited from this method [1]. It provides a more comfortable processing environment and converts intact proteins into amino acids and small peptides, increasing the nutritional value. Additionally, proteolytic enzymes enable selective hydrolysis to create materials that could be safer and more precise. By choosing particular enzymes and specifying the reaction conditions, the functionality of the end products may be managed and scaled up [2]. Protein hydrolysates are frequently utilized as functional additives, flavor enhancers, and nutritional supplements in foods, cosmetics, and fortifying soft drinks and juices [3]. Protein hydrolysate production could be a promising research direction for renewable biological resource utility.

Bioactive peptides (BP) are protein fragments produced by hydrolyzing protein-rich sources with nutritional benefits and positive health impacts [2]. They are generally inactive in their parent protein sequence and less than 10 kDa in molecular weight (Mw), but they can be liberated by enzymatic hydrolysis [4]. Intestinal membranes provide more significant BP absorption opportunities since they are more resistant to digestive system breakdown [5]. Additionally, peptides with antioxidant, anticancer, hypolipidemic, antihypertensive, antimicrobial, and immunity-boosting characteristics that are produced from the enzymatic hydrolysates of protein-rich substances might have good health benefits [3,6,7].

Sesame (*Sesamum indicum* L.) is a widely cultivated crop for cooking and producing edible oil. Sesame seeds offer a high nutritional value with up to 50% oil content and significant levels of minerals, vitamins, and omega-3 fatty acids. The oil has a higher percentage of unsaponifiable matter (2%) than other vegetable oils, including triterpene alcohols, phytosterols, tocopherols, and lignans (primarily sesamin and sesamolin). This increases its oxidative stability and provides several advantageous physiological effects [8]. Sesame cake is a by-product of the oil extraction process, and depending on the method used, its protein concentration can exceed 50% [9].

Due to the high amount and high quality of its protein, sesame cake can be a renewable biological resource to manufacture protein hydrolysates and bioactive peptides [10]. A few experiments on the hydrolysis of sesame cake examined how four different protease types affected the functional characteristics of protein hydrolysates from defatted sesame cake [11]. Research has been done on how process factors affect the hydrolysis of sesame cake hydrolysate [12]. Recently, the antioxidant and antihypertensive properties of white sesame seed protein meal, ultrafiltration membrane peptide fractions, isolate, and enzymatic hydrolysate has been investigated [3,12]. However, the protein hydrolysate from black sesame cake has not yet been documented for its bioactivities.

Consequently, the objective of the current study was to assess the bioactivities of protein hydrolysates and peptide fractions that were separated from defatted black sesame cake. Flavourzyme, a high-quality blend of endo- and exo-peptidases, was used to produce the hydrolysates. After fractionation, in vitro antioxidant, anti-diabetic, anti-obesity, and antihypertensive assays were conducted accordingly.

## 2. Results and Discussion

### 2.1. Enzyme Hydrolysis of DBSC and Ultrafiltration Peptide Fractions

Flavourzyme is a commercial enzyme offered as a peptidase preparation from *Aspergillus oryzae* strain ATCC 42149/RIB 40 (yellow koji mold). Both industrial and academic sectors frequently use this type of enzymatic protein hydrolysis. Among its eight key enzyme activities are two aminopeptidases, two dipeptidyl peptidases, three endopeptidases, and one α-amylase [13]. Utilization of Flavourzyme and other proteolytic enzymes for preparing sesame cake hydrolysates has been reported. Even the greatest level of hydrolysis (DH) was discovered in the alcalase 2.4 L reaction, which Flavourzym immediately followed. However, Flavourzyme showed the greatest protein recovery at 50 °C pH 7.0 [12]. In this work, defatted black sesame cake (DBSC), a by-product of the sesame oil industrial, was hydrolyzed by Flavourzyme for 30, 60, 120, 240, and 360 min, and five hydrolysates were obtained. The estimation of DH monitored digestion efficiency. Figure 1A shows that the DH of the obtained hydrolysates rapidly increased exponentially with time and reached the highest DH value of 26.21% at 60 min. After that, the DH of the hydrolysates collected at 60, 120, 240, and 360 min did not change significantly. This result suggested that the process had reached a stationary phase. The hydrolysis pattern was similar to the previously reported effect [11]. This work’s DH value was higher than previously reported [14] for sesame cake hydrolysates obtained after incubation with alcalase for 1 h (DH of 26.3%). SDS-PAGE analysis was used to establish that DBSC was susceptible to enzymatic hydrolysis. Figure 1B illustrates how hydrolysis steadily reduced the strength of each band that was discernible in the non-hydrolyzed material. However, hydrolysis left an 18 and 30 kDa band largely intact, suggesting that these proteins are resistant to the ability of this commercial enzyme. The majority of sesame seed protein was 11S globulin protein, according to the isolation and protein identification of a storage protein from sesame seeds. SDS-PAGE, an analysis of the protein samples indicated that the separated 11S globulin protein was composed of an acidic polypeptide (30–33 kDa) and a single basic polypeptide (20–24 kDa) [15].

According to their Mw, there have been reports of increased bioactivity of the protein hydrolysates following peptide fractionation [3,4,12]. Processing costs are increased by adding a purification stage. Still, on an industrial scale, the increased economic worth of the product due to its higher bioactivity can make the extra cost justifiable. Another advantage of fractionating hydrolysates is the increased digestion resistance and bioavailability of low-Mw peptides compared to peptides with a more complex Mw distribution [5]. Due to these factors, the BSC was processed using Flavourzyme hydrolysis and then ultrafiltration to separate the peptides into the three groups (<3, 3–10, and >10 kDa), as indicated in the schematic diagram (Figure 2). The separation of SPH by membrane ultrafiltration revealed that the >10 kDa fraction was found in a more significant proportion (65%), followed by 3–10 kDa (20%) and <3 kDa (15%), respectively. Low specificity or resistance to the enzyme activity might result in low efficiency of enzymatic hydrolysis.

### 2.2. Antioxidant Activities of SPH and Peptide Fractions

Academics, consumers, and the food industry are focusing their attention on antioxidant peptides as one essential functional ingredient with potential applications in producing nutritious foods and preserving food quality and safety [16]. After being liberated from their parent protein, the amino acid content, hydrophobicity, sequence, and Mw of peptides all affect how bioactive they are [17]. In peptides containing 2–20 amino acids and Mw of less than 3 kDa, hydrophobic amino acids such as proline, valine, tryptophan, and phenylalanine show high antioxidant action [18]. Due to the potential for differences among distinct radical systems, many studies advised using at least two approaches to evaluate a compound’s antioxidant capacity. The current work used the DPPH, ABTS, and FRAP assays to measure the antioxidant ability of peptide fractions made from SPH hydrolysate. While the FRAP assay measured ferric-reducing activity, the DPPH and ABTS assays measured free radical scavenging capacity.

Figure 3 compares the antioxidant activity of SPH and its peptide components to the standard antioxidant Trolox. The peptide fractions with various molecular weights illustrated in Figure 3A had DPPH radical scavenging activities ranging from 46.72 to 80.67%. The antioxidant capacities were moderately higher than the Trolox (54.14%). However, the DPPH scavenging activity observed herein was lower than the range (77–87%) reported in prior research [12]. The highest DPPH activity was seen in the <3 kDa peptide fraction, whereas the lowest activity was in the SPH (46.72%). The activity of SPH and ultrafiltration membrane peptide-derived fractions to scavenge ABTS radicals is shown in Figure 3B. According to the findings, the <3 kDa peptide fraction demonstrated the strongest activity (58.54%) compared to the larger molecular weight fractions in the range of 38.43 to 55.43% [12]. The relationship between the molecular weight of peptides and their ability to scavenge radicals has been the subject of numerous investigations. Even though high-Mw peptide fractions have repeatedly been shown to be more effective at scavenging radicals than low-Mw fractions [19], other researchers demonstrated that low-Mw peptide fractions have a stronger radical scavenging activity [20]. Additionally, several researchers discovered a J- or a U-shaped association between the radical scavenging activity and the peptide Mw [21]. The various peptide sources and structures could be the reason for the discrepancy in these results [22].

To assess an antioxidant’s capacity to contribute an electron or hydrogen, the ferric reducing antioxidant power (FRAP) assay was used. As shown in Figure 3C, Trolox (1.657) had the most potent reducing power, which was significantly higher than that of SPH (0.843) and peptide fractions (0.438–0.658). The SPH exhibited strong FRAP values compared to its peptide fractions. Furthermore, the reducing power of the three peptide fractions was discovered to increase with the peptide Mw in this study. This outcome was consistent with earlier research on the hydrolysate of white sesame cake and its ultrafiltration peptide fractions [12]. However, in addition to affecting Mw, the type and order of amino acids in the peptide chain may also affect FRAP. The literature has examined the ability of purified peptides to act as antioxidants. The results demonstrated a direct relationship between the peptides’ antioxidant capacity and structural characteristics such as molecular mass, amino acid composition, amino acid sequence, and hydrophobicity. According to previous findings, amino acids, including lysine, leucine, isoleucine, and histidine, may have affected the reducing power activities of protein hydrolysates or peptide fractions [17,23]. The present study supported that the Mw or size of the peptides may influence the antioxidant power of sesame cake peptides. This is owing to the increased availability of hydrogen ions (protons and electrons) due to peptide release via enzymatic hydrolysis.

### 2.3. Anti-Diabetic Properties of Peptide Fractions

Dipeptidyl-peptidase IV (DPP-IV) is a ubiquitous enzyme that acts on incretin hormones, mainly glucagon-like peptide-1 (GLP-1) and gastric inhibitory peptide (GIP), which regulate blood glucose levels through multiple mechanisms. Inhibition of DPP-IV has proven to enhance the body’s own ability to control blood glucose by increasing the activity levels of incretin hormones in the body. Up to now, DPP-IV inhibitors are a group of oral diabetic medications approved by the Food and Drug Administration (FDA) to treat type 2 diabetes mellitus [24,25].

The DPP-IV inhibitory actions of peptides of sizes <3, 3–10, and >10 kDa were derived from DBSC by hydrolysis with Flavourzyme, and ultrafiltration, respectively, were examined in this study. As indicated in Table 1, the <3 kDa peptide fractions showed the highest DPP-IV relative inhibitory ability (IC_50_ = 0.78 mg/mL), followed by 3–10 kDa (IC_50_ = 0.98 mg/mL) and >10 kDa (IC_50_ = 1.34 mg/mL). Numerous investigations have demonstrated that short-sized peptide fractions have a more significant DPP-IV inhibiting effect [26,27]. According to the findings of this in vitro investigation, sesame cake peptides may have the capacity to regulate hyperglycemia.

### 2.4. Anti-Obesity Properties of Peptide Fractions

The activities of α-amylase, α-glucosidase, and pancreatic lipase have been discovered to be inhibited in vitro by several plant proteins and peptides [2,7]. The primary digestive enzymes responsible for breaking down dietary starch are α-amylase and α-glucosidase, which release oligosaccharides that may subsequently be broken down into glucose, quickly absorbed by the body. Thus, inhibiting these enzymes causes a decrease in the sugar absorption rate and benefits in managing both body weight and blood sugar [28]. Aside from hydrolyzing triglycerides into fatty acids and glycerol, pancreatic lipase also produces mono- and diglycerides. Alkaline circumstances are best for the activity of pancreatic lipase. Inhibiting pancreatic lipase reduces the efficiency of fat absorption in the small intestine, which results in a moderate, long-lasting reduction in body weight [29]. Moreover, weight reduction is also responsible for insulin resistance amelioration, which induces an increase in body weight loss in both diabetic and normoglycemic patients, also ameliorating the liver and enhancing muscle metabolism [30]. The anti-obesity effects of peptide fractions were identified by assessing the inhibition of the metabolic enzymes α-amylase, α-glucosidase, and pancreatic lipase. The dose-response curves for three peptide fractions and acarbose, a positive control, are displayed in Figure 4. The α-amylase inhibitory activity was nearly maximal at a 10 mg/mL sample concentration. Acarbose, a commercial-amylase inhibitor, showed more inhibitory activity (92.53%) than peptide fractions (20.12–31.08%). The findings revealed that the 3 kDa fraction (31.08%) had the most inhibitory activity. Comparable to the result found for sesame seed protein hydrolysate previously published by [31], the >10 kDa fraction (20.12%) showed the lowest inhibitory efficacy. Figure 4B demonstrates the ability of sesame peptide fractions to inhibit α-glucosidase. The findings revealed that, compared to standard acarbose, the peptide samples (29.45–35.78%) displayed modestly inhibitory actions (78.23%). Among the peptide fractions, the <3 kDa fraction (35.78%) demonstrated the most significant α-glucosidase inhibition, while the 3–10 kDa fraction (29.45%) exhibited the least inhibitory action. On the other hand, the pancreatic lipase-inhibitory activity increased with increasing peptide Mw. The pancreatic lipase-inhibitory activity of the larger Mw peptide fraction, >10 kDa fraction (57.25%), was substantially (*p* < 0.05) higher than that of the 3–10 kDa (53.21%) and 3 kDa fractions (34.33%). In contrast, Zimmex, positive control, had the highest inhibitory activity (75.67%). Therefore, it is recommended that the peptide part of BSC be incorporated into the composition of products to combat obesity.

### 2.5. Antihypertensive Properties of Peptide Fractions

Angiotensin-Converting Enzyme (ACE) is an enzyme that catalyzes the breakdown of angiotensin I to generate angiotensin II with strong vasoconstrictor activity and the cleavage of the vasodilator bradykinin, hence driving the rise in blood pressure [6]. As a result, ACE inhibitory peptides are thought to be a helpful method for avoiding hypertension. As a result, ACE inhibitory peptides are a useful method for avoiding hypertension. According to Table 1, the peptide fraction with Mw < 3 kDa showed higher ACE inhibitory activity (IC_50_ = 0.15 mg/mL) than 3–10 kDa (IC_50_ = 0.27 mg/mL), and >10 kDa (IC_50_ = 0.37 mg/mL) fractions were observed compared to Alacepril (IC_50_ = 0.001 mg/mL). The findings are consistent with prior research that linked enhanced ACE-inhibitory potentials of low molecular weight peptides to increased ACE active site penetration [32]. Six very short peptides with potent ACE inhibitory action were identified from sesame powder [33]. Ultrafiltration was used to obtain ACE inhibitory peptide, a tiny peptide with Mw of less than 3 kDa, isolated from sesame skin following treatments with papain, pepsin, and alkalize [34]. In addition, three sesame peptide fractions with antihypertensive effects were found to have molecular weights of less than 3 kDa, 3–10 kDa, and larger than 10 kDa [3]. These results showed that Mw interdependency affects the function–activity connection of the anti-hypertension peptide. Overall, according to the findings, the peptide fractions of sesame cake might be used as an ACE inhibitor in living systems and further preclinical and clinical studies are needed.

## 3. Materials and Methods

### 3.1. Materials

Defatted black sesame cake (DBSC) was used as the protein source, and it was produced industrially using a cold press oil machine from Khon Pong Yung Mha Community Enterprise (Chiang Mai, Thailand). All enzymes and substrates for in vitro inhibitory assay were obtained from Sigma (St. Louis, MO, USA). Other reagents of analytical quality were purchased from Merck (Darmstadt, Germany), Fluka (Buchs, Switzerland), or Fischer (Schwerte, Germany).

### 3.2. Preparation of DBSC Protein Hydrolysate (SPH) and Peptide Fractions

The DBSC (5% *w*/*v*) was hydrolyzed at 50 °C with 1 % (*w*/*w*) of Flavourzyme after being suspended in 200 mL of 0.1 M phosphate buffer (pH 7.0) (Novo Nordisk Co., Bagsværd, Denmark). The enzymatic reaction was stopped by inactivating the enzyme for 10 min at 95 °C. The mixture was centrifuged for 30 min at 10,000× *g* at 4 °C after being cooled to room temperature. The final supernatant was successively passed through ultrafiltration membranes with molecular weight cut-off (MWCO) of 3 and 10 kDa in an Amicon Ultra Centrifugal filter (Millipore, Massachusetts, USA). Three fractions of peptides (<3 kDa, 3–10 kDa, and >10 kDa) were separated from the permeate of each MW cut-off membrane. The three oligopeptide fractions and SPH were freeze-dried and kept at −20 °C until further use to facilitate analysis.

### 3.3. DPPH radical Scavenging Assay

Free radical scavenging ability was determined by DPPH radical scavenging assay [35], with slight modification. In brief, 100 µL of (1 mg/mL) SPH and three oligopeptide fractions were mixed with 100 µL of 0.4 mM DPPH solution and incubated in the dark for 30 min at room temperature. A control sample was performed using distilled water. The positive control was Trolox (1 mg/mL), an analog of vitamin E that is water soluble. Next, a microplate reader was used to measure the absorbance at a 517 nm wavelength (BioTek Instruments Inc., Winooski, VT, USA). The following equation was used to calculate the ability of DPPH radical scavenging and express it as a percentage inhibition: DPPH inhibition (%) = [(A_0_ − A_S_)/A_0_] × 100, where A_0_ is the absorbance of the control, and A_S_ is the absorbance of the sample or the positive control.

### 3.4. ABTS^•+^ Radical Scavenging Assay

Radical scavenging activity (ABTS^•+^) was assessed using the technique previously described [36]. As stock solutions, 14 mM of ABTS solution and 4.9 mM of potassium persulfate solution were prepared. The two stock solutions were mixed in a ratio of 1:1 (*v*/*v*) to obtain the working ABTS^•+^ solution, which was then left to react for 12 h at room temperature in the dark. A spectrophotometer was used to measure the absorbance of the solution at 734 nm, which derived 0.700 ± 0.02 units after being diluted with distilled water. A mixture of 3.0 mL of the ABTS^•+^ solution and 1 mL of distilled water containing 1 mg/mL of the SPH and three oligopeptide fractions was added. This mixture was then incubated at room temperature for 3 min in the dark. Distilled water was used for the control sample. One milligram per milliliter of Trolox was used as the positive control. A UV/Vis spectrophotometer was used to measure the absorbance of the solution at 734 nm (Shimadzu, Kyoto, Japan). The following equation was used to calculate the percentage of ABTS^•+^ radical scavenging inhibition: ABTS^•+^ inhibition (%) = [(A_0_ − A_S_)/A_0_] × 100, where A_0_ is the absorbance of the control, and A_S_ is the absorbance of the sample or the positive control.

### 3.5. Ferric Reducing Antioxidant Power (FRAP) Assay

The ferric-reducing antioxidant power (FRAP) was assessed using a slightly modified version of the FRAP assay described in a prior study. In brief, the FRAP reagent was freshly achieved by combining 10 mM 2,4,6 tripyridyl-s-triazine (TPTZ) in 40 mM HCl, 20 mM FeCl_3_, and 300 mM acetate buffer (pH 3.6) in the ratio 10:1:1 (*v*/*v*/*v*). In this experiment, 180 µL of the FRAP reagent and 20 µL of the samples were added to 96-well plates and incubated for 5 min at room temperature in the dark to complete the reaction. A microplate reader was used to measure the reacting samples at 595 nm. As a positive control, Trolox (1 mg/mL) was utilized. Instead of the sample, distilled water was used to prepare the blank control. An increase in absorbance at 700 nm indicated that protein hydrolysates had more reducing power.

### 3.6. Inhibition of the α-Glucosidase Activity

A modified approach was used to execute the suppression of α-glucosidase activity protocol [37]. A 96-well plate was filled with 50 µL of 1 mM p-nitrophenyl-D-glucopyranoside (p-NPG) and 50 µL of the sample solution. After 5 min incubation at 37 °C, 10 µL of buffer containing 0.01 U/mL of α-glucosidase from Saccharomyces cerevisiae Type I (G5003) was added to each well. The reaction was then incubated once more for 30 min at 37 °C. After incubation, the process was stopped by adding 100 µL of 1 M sodium carbonate solution. Acarbose (PHR1253) was used as the positive control. A microplate reader was used to measure the reaction mixture’s absorbance at 400 nm. The percentage of enzyme inhibitory action was computed using the following formula: Alpha-glucosidase inhibition (%) = [(A_NC_ − A_S_)/A_NC_] × 100, where A_NC_ is the absorbance of the control (no inhibitor), and A_S_ is the absorbance of the sample or the positive control.

### 3.7. Inhibition of the α-Amylase Activity

The α-amylase inhibitory activity of samples was measured using a slightly modified approach as previously described [28]. Pre-incubation at 37 °C for five minutes was done on a reaction mixture containing 50 µL of sample solution, 10 µL of phosphate buffer (100 mM, pH 6.9), and 20 µL of (0.1 U) porcine pancreas-amylase (A3176). Next, 20 µL of 1% soluble starch was added as the substrate, and the mixture was incubated at 37 °C for an additional 5 min. After adding 50 µL of 1 M HCl and 50 µL of iodine solution, the process was stopped. As a positive control, acarbose was utilized. A microplate reader was used to measure the absorbances at 650 nm. The following formula was used to calculate the percentage of enzyme inhibitory action: Alpha-amylase inhibition (%) = 100 − [(A_0_ − A_s_) × 100/(A_0_ − A_NC_)], where A_0_ is the absorbance of the blank (without enzyme), A_NC_ is the absorbance of the control (without inhibitor), A_S_ is the absorbance of the sample or the positive control.

### 3.8. Inhibition of Pancreatic Lipase Activity

An in vitro inhibition of pancreatic lipase activity was used to determine the anti-hyperlipidemia assay as described in a prior study with slight modification [38]. A reaction mixture composed of 50 µL of various sample concentrations, 2.5 µL of 0.05 M 4-Nitrophenyl butyrate (p-NPB), and 137.5 µL of 100 mM phosphate buffer pH 7.0 was prepared. Before adding 10 µL of Type II Lipase from the porcine pancreas, the reaction mixture was pre-incubated at 37 °C for 5 min. After that, the reaction was kept at 37 °C for an additional 30 min. Zimmer was used as a positive control. The absorbances at 410 nm were measured using a microplate reader. The percentage of enzyme inhibitory action was computed using the following formula: pancreatic lipase inhibition (%) = [(A_NC_ − A_S_)/A_NC_] × 100, where A_NC_ is the absorbance of the control (no inhibitor), and A_S_ is the absorbance of the sample or positive control.

### 3.9. Inhibition of Angiotensin-Converting Enzyme (ACE)

The ACE-inhibitory activities of the different sample concentrations were investigated using an ACE kit-WST (Dojindo Laboratories, Kumamoto, Japan). The assay was carried out following the manufacturer’s instructions. A microplate reader was used to measure the reactions’ absorbances at 450 nm. The following equation calculated the ACE-inhibitory activities of the samples: ACE-inhibitory activity (%) = [(A_blank1_ − A_sample_)/(A_blank1_ − A_blank2_)] × 100, where A_blank1_ is the absorbance of the positive control (without ACE inhibition), A_sample_ is the absorbance of the sample, and A_blank2_ is the absorbance of the reagent blank. The IC_50_ value was determined using nonlinear regression analysis. To identify statistical differences, the one-tailed Welch’s *t*-test was used. *p*-values < 0.05 were regarded as significant. The information was presented as means of the IC_50_ value.

### 3.10. Inhibition of Dipeptidyl Peptidase IV (DPP-IV)

The DPP-IV inhibition screening kit (Catalog Number MAK203, Sigma-Aldrich, Missouri, Germany) was used to assess the DPP-IV inhibitory effect of the oligopeptide fractions. The assay was carried out following the manufacturer’s instructions. The cleavage of the substrate to produce a fluorescent product is the basis for measuring the DPP-IV activity (λex = 360 nm/λem = 460 nm), depending on the amount of enzymatic activity in the test sample. A standard DPP-IV inhibitor (Sitagliptin), which was included with the kit, was used to compare the efficiency of the tested inhibitors. The measurement was deployed with kinetic mode (reading every minute) for 15–30 min at 37 °C, and the DPP-IV inhibition activity of the samples was given as a percentage of relative inhibition (relative inhibition activity; RIA) to the manufacturer’s protocol. The results expressed the IC_50_ value or sample concentration needed to inhibit 50% DPP-IV activity.

### 3.11. Statistical Analysis

The mean and standard deviation of triple experiments expresses the results. SPSS 21.0 (IBM, Armonk, NY, USA) was used for the multivariate analysis of variance. Duncan’s multiple range test was used to find significant discrepancies between mean values. *p* < 0.05 was used as the criterion of significance.

## 4. Conclusions

According to our research, an evaluation of the antioxidant, anti-diabetic, anti-obesity, and antihypertensive properties of defatted sesame seed cake protein hydrolysate and its ultrafiltered peptide fractions was conducted. The sesame cake products’ abilities to scavenge free radicals and inhibit the redox reaction of ferric ions were variable, which may be explained by changes in their size, solubility, or content of amino acids. The low Mw peptide fraction (<3 kDa) showed the highest α-amylase, α-glucosidase, ACE, and DPP-IV inhibitory activities, while larger peptides (>3 kDa) exhibited higher inhibition against pancreatic lipase. According to the findings, sesame hydrolysate and its peptides could be used as ingredients in the treatment of free radical-related disorders, the prevention of atherosclerosis in blood vessels, and the inhibition of various enzymes involved in the progression of hypertension and hyperglycemia. The in vitro qualities mentioned in this paper must be confirmed in subsequent in vivo investigations utilizing animal models.

## Figures and Tables

**Figure 1 molecules-28-00211-f001:**
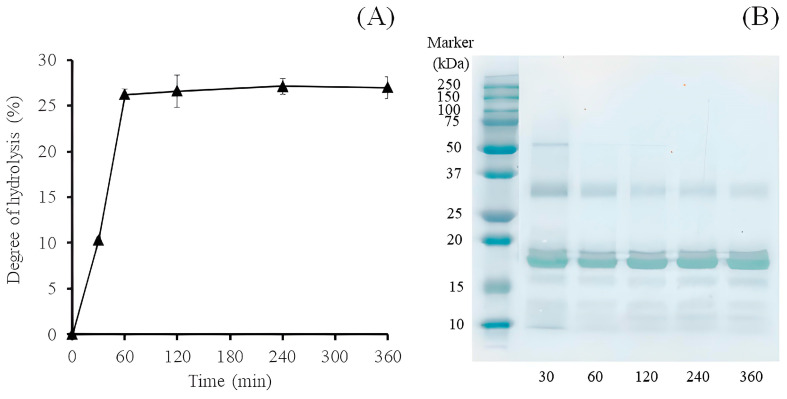
Degree of hydrolysis (%) of the hydrolysates from DBSC after the action of Flavourzyme for 360 min (**A**). SDS-PAGE analysis of SPH with Flavourzyme (**B**).

**Figure 2 molecules-28-00211-f002:**
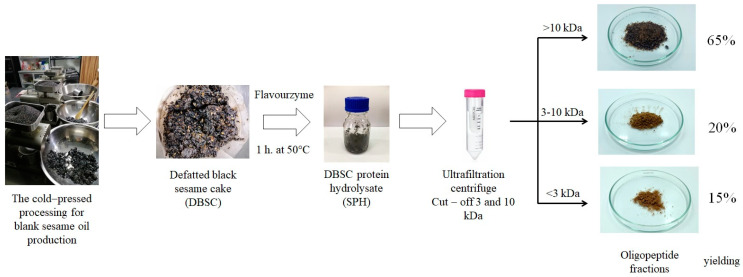
The processing of oligopeptide fractions from SPH.

**Figure 3 molecules-28-00211-f003:**
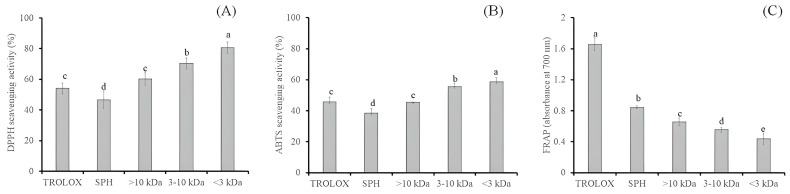
Antioxidant activities of Oligopeptide fractions. DPPH scavenging activity (**A**), ABTS scavenging activity (**B**), and Ferric (Fe^3+^) reducing power (**C**). Data expressed as mean ± SD, (*n* = 3). Bars with different alphabets are significantly different at *p* < 0.05.

**Figure 4 molecules-28-00211-f004:**
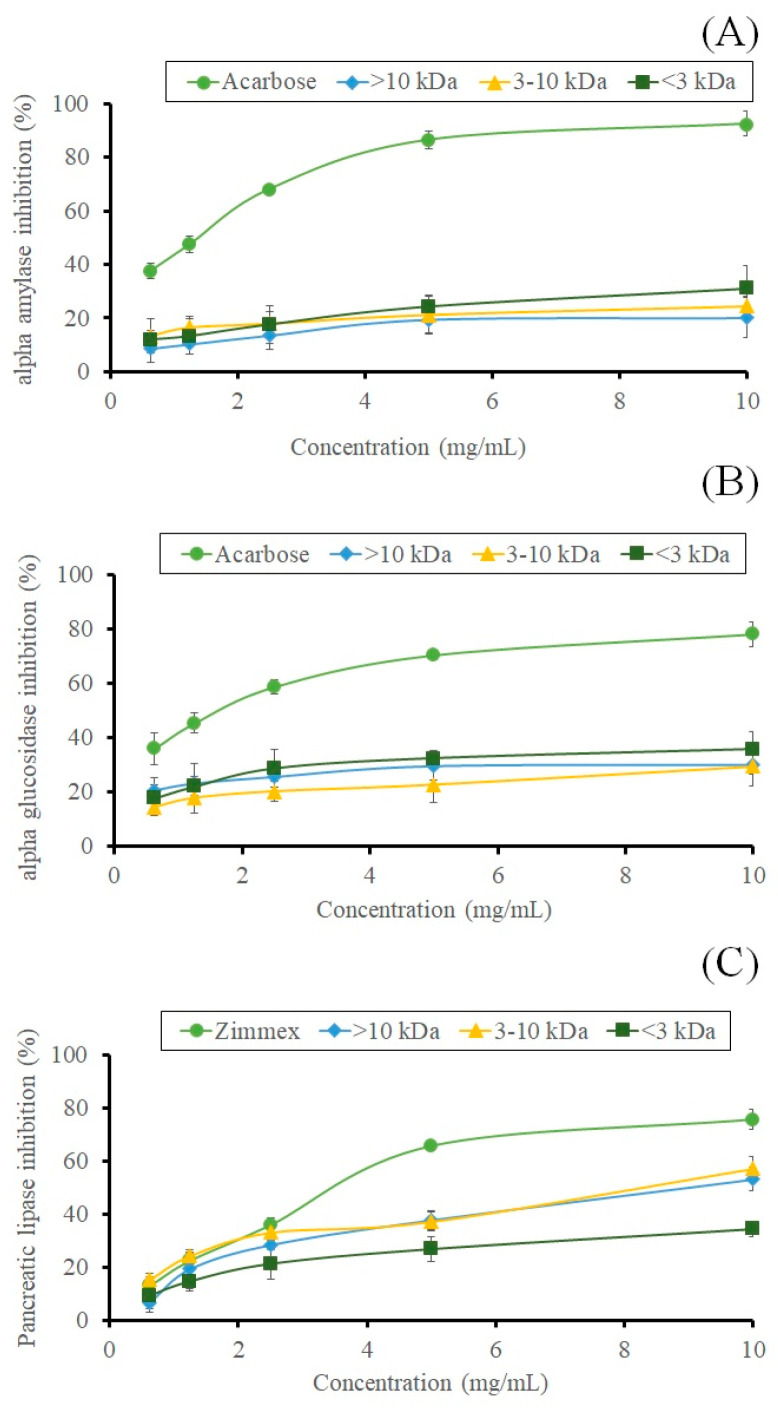
Anti–obesity effects of oligopeptide fractions. Inhibition of α–amylase (**A**), inhibition of α–glucosidase (**B**), and inhibition of pancreatic lipase (**C**) enzyme. Data expressed as mean ± SD, (*n* = 3). Bars with different alphabets are significantly different at *p* < 0.05.

**Table 1 molecules-28-00211-t001:** Inhibitory activity (expressed as IC_50_ or oligopeptide concentration needed to inhibit the original enzyme activity by 50%) of oligopeptide fractions by ultrafiltration against angiotensin-converting enzyme (ACE) and dipeptidyl peptidase-IV (DPP-IV).

Enzymatic Inhibitory Activity (IC_50_, mg/mL) *		
Samples	ACE	DPP-IV
Positive control	0.001 ± 0.000 ^a^	0.003 ± 0.000 ^a^
>10 kDa	0.37 ± 0.04 ^b^	1.34 ± 0.12 ^b^
3–10 kDa	0.27 ± 0.02 ^c^	0.98 ± 0.06 ^c^
<3 kDa	0.15 ± 0.03 ^d^	0.78 ± 0.11 ^d^

* Values are the mean ± standard deviation of three independent experiments. Different lowercase letters indicate significant differences among samples (*p* < 0.05). Positive control was Alacepril in the ACE inhibitory assay and Sitagliptin in the DPP-IV inhibitory assay.

## Data Availability

Not applicable.

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
