# Peer review of "Antioxidant, Anti-Diabetic, Anti-Obesity, and Antihypertensive Properties of Protein Hydrolysate and Peptide Fractions from Black Sesame Cake"

_molecules, 2022, doi:10.3390/molecules28010211_

Round 1

Reviewer 1 Report

The manuscript “Antioxidant, anti-diabetic, anti-obesity, and antihypertensive properties of protein hydrolysate and peptide fractions from black sesame cake” submitted to Molecules journal represents an interesting study as regard by-products utilization is a hot trend. The paper deal with defatted black sesame cake and in particular with remaining protein and essential amino acids that may be utilized as a renewable biological source to produce bioactive products. The bioactivities of the protein hydrolysate from black sesame cake and its peptide fractions are the focus of the study.

The research design is appropriate and the results are valuable.

I have the following minor comments/questions to the authors in order to improve the manuscript:

- Pay more attention to the formatting of tables footers

- Use either DPP4 or DPP-IV – please unify the whole text

- Please be more precise in DPPH writing. What is the difference between the two radical assays and the reason to write once DPPH radical scavenging activity and then ABTS•+ radical scavenging assay? Please use the same approach.

- Use italics for in vivo and in vitro

- What about health concerns in using protein hydrolysates?

- What is the origin of the defatted black sesame cake? Please mention this in the text.

- At what concentration Trolox was used as a control?

Author Response

Dear Reviewer

Thank you very much for reviewing our manuscript titled “Antioxidant, anti-diabetic, anti-obesity, and anti-hypertensive properties of protein hydrolysate and peptide fractions from black sesame cake”. The edited version of our manuscript with the following changes was done according to the editor's and reviewers’ comments. The changes were presented with “Track Changes” in MS Word.

Responses to editor's and reviewers’ comments

From the following information in the revised Manuscript ID: molecules-2126241

Reviewer 1

  1. Pay more attention to the formatting of tables footers

            The tables footers were changed in font size from 10 pt to 9 pt refer to the template.

  1. Use either DPP4 or DPP-IV – please unify the whole text

            We have changed to use DPP-IV for the whole text.

  1. Please be more precise in DPPH writing. What is the difference between the two radical assays and the reason to write once DPPH radical scavenging activity and then ABTS•+ radical scavenging assay? Please use the same approach.

            The DPPH and ABTS•+ radical scavenging assays employ the same principle. A synthetic colored radical or redox-active compound is generated; and the ability of a biological sample to scavenge the radical is monitored by a spectrophotometer, applying an appropriate standard to quantify antioxidant capacities such as Trolox or vitamin C. In the specification, the ABTS assay is based on the generation of a blue/green ABTS•+ that can be reduced by antioxidants; whereas the DPPH assay is based on the reduction of the purple DPPH to 1,1-diphenyl-2-picryl hydrazine. However, both assays are convenient in their application and thus most popular. For your suggestion, we already changed the writing of “DPPH radical scavenging activity” to “DPPH radical scavenging assay” for the same approach as ABTS•+ radical scavenging assay.

  1. Use italics for in vivo and in vitro

            We have changed in vivo and in vitro to the italics in the whole text.

  1. What about health concerns in using protein hydrolysates?

            Similar to the intake of intact proteins with a history of safe use, the intake of hydrolysates made from them, does not raise safety concerns.

  1. What is the origin of the defatted black sesame cake? Please mention this in the text.

            Defatted black sesame cake (DBSC) is a by-product of the sesame oil industry, which is produced from a cold press oil machine or a hydraulic screw press oil machine. Our DBSC was carried out by Khon Pong Yung Mha Community Enterprise, Chiang Mai, Thailand. This information has mentioned in Lines 81-82 and Lines 265-267.

  1. At what concentration Trolox was used as a control?

            1 mg/mL or 4 mM of Trolox was used as the positive control in this work.

Reviewer 2 Report

I read with great interest the paper “Antioxidant, anti-diabetic, anti-obesity, and antihypertensive properties of protein hydrolysate and peptide fractions from black sesame cake” by Chaipoot et al.

The design is fine. The article is logically divided into sections and subsections. Data presented are of good interest.

Comments:

1.      Line 175-181: It has already been established the role od DDP4 inhibitors in type 2 diabetes mellitus treatment. They have proven a good efficacy in serum glucose reduction, though, up to now, have proven low effects on cardiovascular risk reduction when compared to other drugs (doi: 10.1001/jama.2019.11489).

2.      α-amylase and α-glucosidase as reported by the authors reduce the absorption of both glucose and lipids, thus inducing a weight reduction. To this mechanism it is also important to add another one. In fact, weight reduction is responsible of insulin resistance amelioration, which induces an increase in body weight loss in both diabetic and normoglycemic patients, also ameliorating the liver and enhancing the muscle metabolism (doi: 10.37349/emed.2020.00019).

3.      Line 254-255: It is possible that this peptides in future could be taken into consideration as ACEi. However, one of the most important issues to raise is that by inhibiting ACE enzyme, there would be a shift to bradykinin, which, beyond the vasodilatory effect, may also reproduce cough and other side effect, which, for now represent a need for drug discontinuation. Thus, I suggest that could should be changed with might, and also to add that further preclinical and clinical studies are needed.

Author Response

Dear Reviewer

Thank you very much for reviewing our manuscript titled “Antioxidant, anti-diabetic, anti-obesity, and anti-hypertensive properties of protein hydrolysate and peptide fractions from black sesame cake”. The edited version of our manuscript with the following changes was done according to the editor's and reviewers’ comments. The changes were presented with “Track Changes” in MS Word.

Responses to editor's and reviewers’ comments

From the following information in the revised Manuscript ID: molecules-2126241

Reviewer 2

  1. Line 175-181: It has already been established the role of DDP4 inhibitors in type 2 diabetes mellitus treatment. They have proven a good efficacy in serum glucose reduction, though, up to now, have proven low effects on cardiovascular risk reduction when compared to other drugs (doi: 10.1001/jama.2019.11489).

            Lines 178-184, we have re-described the recent information on the DPP-IV enzyme and DPP-IV inhibitor as your suggestion.

  1. α-amylase and α-glucosidase as reported by the authors reduce the absorption of both glucose and lipids, thus inducing a weight reduction. To this mechanism it is also important to add another one. In fact, weight reduction is responsible of insulin resistance amelioration, which induces an increase in body weight loss in both diabetic and normoglycemic patients, also ameliorating the liver and enhancing the muscle metabolism (doi: 10.37349/emed.2020.00019).

            Line 213-215, we have added more information as your suggestion.

  1. Line 254-255: It is possible that this peptides in future could be taken into consideration as ACEi. However, one of the most important issues to raise is that by inhibiting ACE enzyme, there would be a shift to bradykinin, which, beyond the vasodilatory effect, may also reproduce cough and other side effect, which, for now represent a need for drug discontinuation. Thus, I suggest that could should be changed with might, and also to add that further preclinical and clinical studies are needed.

               Line 254-255, we have changed “could” to “might”, and also added “that further preclinical and clinical studies are needed” in the sentence.
